# LINC01270 Regulates the NF-κB-Mediated Pro-Inflammatory Response via the miR-326/LDOC1 Axis in THP-1 Cells

**DOI:** 10.3390/cells13232027

**Published:** 2024-12-08

**Authors:** Imene Arab, Su-Geun Lim, Kyoungho Suk, Won-Ha Lee

**Affiliations:** 1School of Life Sciences, Kyungpook National University, Daegu 41566, Republic of Korea; arabimene07@gmail.com (I.A.); sugeun624@hanmail.net (S.-G.L.); 2BK21 FOUR KNU Creative BioResearch Group, Kyungpook National University, Daegu 41566, Republic of Korea; 3Brain Science & Engineering Institute, Kyungpook National University, Daegu 41944, Republic of Korea; 4Department of Pharmacology, School of Medicine, Kyungpook National University, Daegu 41944, Republic of Korea; 5BK21 Plus KNU Biomedical Convergence Program, School of Medicine, Kyungpook National University, Daegu 41944, Republic of Korea

**Keywords:** LINC01270, NF-κB signaling, miR-326, LDOC1, inflammation

## Abstract

Long intergenic noncoding (LINC)01270 is a 2278 bp transcript belonging to the intergenic subset of long noncoding (lnc)RNAs. Despite increased reports of LINC01270’s involvement in different diseases, evident research on its effects on inflammation is yet to be achieved. In the present study, we investigated the potential role of LINC01270 in modulating the inflammatory response in the human monocytic leukemia cell line THP-1. Lipopolysaccharide treatment upregulated LINC01270 expression, and siRNA-mediated suppression of LINC01270 enhanced NF-κB activity and the subsequent production of cytokines IL-6, IL-8, and MCP-1. Interestingly, the knockdown of LINC01270 downregulated expression of leucine zipper downregulated in cancer 1 (LDOC1), a novel NF-κB suppressor. An analysis of the LINC01270/micro-RNA (miRNA)/protein interactome profile identified miR-326 as a possible mediator. Synthetic RNA agents that perturb the interaction among LINC01270, miR-326, and LDOC1 mRNA mitigated the changes caused by LINC01270 knockdown in THP-1 cells. Additionally, a luciferase reporter assay in HEK293 cells further confirmed that LINC01270 knockdown enhances NF-κB activation, while its overexpression has the opposite effect. This study provides insight into LINC01270’s role in modulating inflammatory responses to lipopolysaccharide stimulation in THP-1 cells via the miR-326/LDOC1 axis, which negatively regulates NF-κB activation.

## 1. Introduction

Inflammation is a complex biological process orchestrated by the immune system that allows the body to withstand harmful stimuli such as pathogens and virulence factors [1]. A multitude of cellular and molecular mechanisms are involved in regulating inflammation, among which the transcription factor NF-κB plays a central role [2]. NF-κB occupies a key position in the regulation of multiple facets of the inflammatory response. It stimulates the transcription of numerous pro-inflammatory cytokines and chemokines. Furthermore, NF-κB is implicated in the regulation of inflammasome activation, innate immune cell activation and differentiation, and the modulation of apoptosis and cell survival [3]. The fine-tuning of NF-κB induction during the inflammatory response is essential to restoring homeostasis and avoiding the detrimental effects of chronic inflammation onset. Thus, the regulation of NF-κB activity is considered a main target for the treatment of a variety of diseases in which inflammation plays an essential role [4,5,6].

Recent research has identified various types of non-coding (nc)RNAs as essential regulators of cellular processes, including gene expression and signal [7,8]. Among these, microRNAs (miRNAs) are short RNAs that silence gene expression by binding to complementary mRNAs, leading to their degradation or blocking translation. Long non-coding (lnc)RNAs, which are longer than 200 bases, act as “sponges” for miRNAs, modulating their availability and indirectly influencing gene expression, while also being involved in transcriptional regulation and chromatin remodeling. Similarly, circular (circ)RNAs, which are circular and stable, also function as miRNA sponges, competing for miRNA binding and thus impacting gene expression networks. Lastly, Piwi-interacting (pi)RNAs interact with Piwi proteins, mainly to silence transposons in germ cells and maintain genome stability, although they may also target specific mRNAs and ncRNAs in certain contexts. Together, these ncRNAs contribute to a complex regulatory network influencing cellular processes and gene expression.

Beyond the well-established roles of various proteins as modulators of the inflammatory response in general, and of NF-κB activation particularly, lncRNAs have emerged as components of the regulatory network of several pathological processes through their modulation of NF-κB activity [2]. For example, lncRNA-cox2 promotes inflammasome sensor expression by facilitating NF-κB p65 nuclear translocation and transcription [9]. Recently, the lncRNA LINC01270 has garnered significant attention due to its involvement in the progression of different cancer types. It promotes esophageal cancer progression by recruiting DNA methyltransferase proteins (DNMTs) to induce the methylation of the glutathione S-transferase P1 (GSTP1) gene and, subsequently, resistance to 5-fluorouracil chemotherapy [10]. Moreover, Li et al. [11] highlighted LINC01270’s role in breast cancer development, where it recruits DNMTs to promote the methylation of the laminin subunit alpha-2 (LAMA2) gene. The LAMA2 protein suppresses malignancy by inhibiting mitogen-activated protein kinase (MAPK) signaling. These two studies unveiled the role of LINC01270 in epigenetic regulation, which is a probable outcome of its nuclear localization. Conversely, LINC01270 exhibits competitive endogenous (ce)RNA functions, as seen in its ability to sponge miRNA (miR)-326 to modulate the mRNA levels of LA-related protein 1 (LARP1) and Ephrin A3 (EFNA3), thus exacerbating lung cancer and gastric cancer progression, respectively [12,13].

miR-326 has been extensively studied in the last decade and found to be important in controlling different cellular processes, such as the differentiation and maturation of immune cells, where it plays a remarkable immunomodulatory role [14,15,16]. It is also involved in regulating oncogenesis, cell invasion, migration, and apoptosis [17]. In autoimmune-induced hepatitis, miR-326 inhibition provided beneficial effects by reducing hepatocyte pyroptosis and the release of pro-inflammatory cytokines. Inhibition of miR-326 resulted in the upregulation of tet methylcytosine dioxygenase 2 (TET2), which is known to inhibit NF-κB activation [18]. In contrast, Wang et al. [19] demonstrated that miR-326 has the ability to diminish sepsis-induced lung injury by targeting TLR4, consequently reducing NF-κB-mediated inflammation.

Leucine zipper downregulated in cancer 1 (LDOC1) is one of several proteins that are involved in regulating NF-κB activity. In the BxPC-3 pancreatic cancer cell line, LDOC1 overexpression inhibited NF-κB activation [20]. In Group A ependymoma, LDOC1 silencing due to promoter methylation led to an upregulation of NF-κB and heightened secretion of IL-6 [21]. Moreover, Zhao et al. [22] demonstrated that LDOC1 sensitizes the papillary thyroid carcinoma-derived thyroid cell line TPC-1 to apoptosis and suppresses their proliferation by inhibiting NF-κB activation.

In our study, we focused on exploring the potential role of LINC01270 in modulating inflammatory response in the THP-1 cell line. Through a series of experiments and by analyzing the lncRNA–miRNA–protein interactome profile, we speculate that LINC01270 acts as a regulator of NF-κB activation by alleviating the repression of LDOC1 imposed by miR-326.

## 2. Materials and Methods

### 2.1. Cell Lines and Cell Culture

Human monocytic leukemia cell line THP-1 cells purchased from the American Type Culture Collection (ATCC) were cultured in Roswell Park Memorial Institute (RPMI) 1640 (WELGENE, Daegu, Korea) culture medium supplemented with 10% FBS, 0.05 mM β-mercaptoethanol, 25 µg/mL glucose, pen-streptomycin, 1-mM sodium pyruvate, and 10 Mm HEPES. Human embryonic kidney cells HEK293 (Korean Cell Line Bank, Seoul, Korea) were cultured in a high glucose Dulbecco’s modified Eagle’s medium (DMEM) (WELGENE) culture medium supplemented with 10% FBS and pen-streptomycin. Cell cultures were maintained at 37 °C in the presence of 5% CO_2_.

### 2.2. RNA Isolation and qRT-PCR

To analyze lncRNA and mRNA expression, total RNA was first extracted by the phenol-chloroform method using TRIzol (Bioscience Technology, Daegu, Korea). For cDNA synthesis, RNA was incubated for 1 h at 45 °C with 5× Reverse-Transcriptase mix (Elpis Biotech, Daejeon, Korea) followed by enzyme inactivation for 10 min at 70 °C. RT-qPCR was performed using SYBR Green Premix (Enzynomics, Daejeon, Republic of Korea) in a qTOWER^3^ thermocycler (Jena Analytik, Jena, Germany) using the primers cited in Table 1. Relative gene expression was calculated using the 2^−ΔΔCt^ method, and β-Actin and GAPDH were used as housekeeping genes. For miR-326 transcript quantification, following TRIzol extraction, reverse transcription and qPCR were performed using miRCURY LNA miRNA PCR Starter Kit (Qiagen, Valencia, CA, USA) following the manufacturer’s instructions, with the U6 gene serving as the endogenous control to normalize miR-326 expression.

### 2.3. ELISA

To quantify protein secretion, THP-1 cells were first seeded at 1 × 10^5^/100 µL in a 96-well plate, and after 24 h of incubation with LPS, the supernatant was collected by centrifugation. ELISA kits for MCP-1 and IL-6, purchased from Invitrogen (Carlsbad, CA, USA), and for IL-8, purchased from Biolegend (San Diego, CA, USA), were used according to the manufacturer’s instructions. Colorimetric changes were measured using a microplate reader (SPECTROstar Nano, BMG LABTECH, Cary, NC, USA) at 450 nm with corrected absorption at 540 nm. Measurements were conducted in duplicate, and the concentration of each sample was calculated in reference to a standard.

### 2.4. Western Blot

The cell pellet was lysed with NP40 lysis buffer supplemented with a protease and phosphatase inhibitor cocktail, sonicated, and incubated for 10 min on ice. After centrifugation at 4 °C and 12,000 rpm for 15 min, the supernatant was collected, mixed with Laemmli buffer supplemented with 100 Mm dithiothreitol, and then boiled for 5 min. Protein samples were separated by sodium dodecyl sulfate-polyacrylamide gel electrophoresis (SDS-PAGE) and then wet transferred into polyvinylidene fluoride. Membranes were blocked in nonfat skimmed milk, or 5% BSA for phosphorylated form proteins, for 1 h and then incubated at 4 °C overnight with the target primary antibody diluted in 5% BSA, as detailed in Table 2. Horseradish peroxidase-labeled secondary antibodies were applied to the membranes, and they were incubated at room temperature for 1 h. Protein bands were visualized using an ECL kit (GenDEPOT, Baker, TX, USA) and a DAVINCH-K Chemi-Fluoro Imager (Seoul, Republic of Korea).

### 2.5. Cell Transfection

To induce LINC01270 attenuation, two sets of siRNA sequences purchased from Bioneer (Daejeon, Korea) were used (Table 3). Briefly, THP-1 cells were seeded at 2 × 10^5^/mL in antibiotic-free RPMI 1640 medium. THP-1 cells were transfected with a mix of siLINC01270-1 and siLINC01270-2 or with scramble serving as a control, at final concentrations of 100 nM using DharmaFECT 1 transfection reagent (Dharmacon, Lafayette, CO, USA) according to the manufacturer’s protocol. Transfection was maintained for 40–48 h before further experimentation. To disrupt the interaction between lncRNA, miRNA, and mRNA, we designed single-stranded RNA oligonucleotides synthesized by Bioneer. These decoy RNAs were designed to mimic the binding sites between the RNA triad (Table 3). Subsequently, we transfected these decoy RNAs into THP-1 cells at a final concentration of 300 nM and incubated them for 48 h. For comparison, we also transfected miR-326 and scrambled inhibitors (Bioneer-Korea) using the same transfection procedures and conditions as used for the decoy RNAs.

### 2.6. dCas9 Overexpression

The overexpression of LINC01270 was achieved using a CRISPR-activation approach. The dCas9-VPR64 plasmid was purchased from Addgene, and crRNA targeting a sequence upstream of the LINC01270 transcription starting site was designed using CRISPR-ERA (http://crispr-era.stanford.edu/, accessed on 22 Novembre 2022). Both synthetic crRNA and TracrRNA were synthetized by Dharmacon, and transfection was executed according to the Dharmacon crispr-a protocol. Briefly, HEK293 cells were pre-seeded at 2 × 10^5^/mL overnight in antibiotic-free DMEM. The CrRNA–TracrRNA complex was formed at room temperature and then transfected at a 200 nM final concentration along with the dCas9-VPR64 plasmid using DharmaFECT 1 transfection reagent (Dharmacon). For the control treatment, the dCas9-VPR64 plasmid without any targeting crRNA was transfected. Western blot analysis was performed by treating HEK293 cells with Recombinant human tumor necrosis factor-α (TNF-α) purchased from PeproTech (Rocky Hill, NJ, USA).

### 2.7. Dual Luciferase Assay

For the luciferase assays, NF-κB or LDOC1 3′UTR luciferase activity was assessed. Briefly, HEK293 cells were seeded in a 96-well plate at 1 × 10^4^/100 µL DMEM overnight with three replicates per sample. Transfection of the appropriate vectors was achieved by resuspending a mix of 300 ng of DNA and Polyfect Transfection Reagent (Qiagen) in antibiotic-free DMEM, adding it to the cell culture, and then incubating the plate for 24 h. After cell lysis, luciferase activity was measured using the Promega DLR system (Madison, WI, USA). The relative activity was normalized to Renilla values in each sample. The LDOC1 3′UTR reporter was purchased from Applied Biological Materials Inc. (Richmond, British Columbia, Canada) (Cat. No. 26423081).

### 2.8. Phagocytosis Assay

Transfected THP-1 cells were seeded at 2 × 10^5^/1 mL complete RPMI medium. *Escherichia coli* (strain K-12) BioParticles (E23370, Invitrogen) were opsonized using an opsonizing reagent (E2870, Invitrogen) at a 1:10 ratio. Cells were treated and incubated for 3 h at 37 °C, washed twice, and then resuspended in 500 µL of Dulbecco’s phosphate-buffered saline. To measure the internalized *E. coli* particle numbers, cell samples were subjected to flow cytometry (BD FACSVerse, BD Biosciences, Franklin Lakes, NJ, USA).

### 2.9. Prediction of LINC01270 Interaction Sites

Bioinformatic tools were employed to identify potential interactions among lncRNA, miRNA, and mRNA. The prediction of LINC01270 potential miRNA targets was conducted using the lncRNASNP database. Additionally, the putative interaction sites between miR-326 and LDOC1 were predicted using the miR-DB and TargetScan databases.

### 2.10. Statistical Analysis

The data are presented as means ± SEM. All experiments were conducted with at least three biological replicates. All statistical tests and data visualizations were performed using GraphPad Prism 5. Statistical significance was assessed using unpaired Student’s *t*-tests when comparing means of two independent groups and two-way ANOVAs followed by Bonferroni post-hoc tests when comparing means of more than two groups. A *p*-value of <0.05 was accepted as indicating statistical significance in all tests.

## 3. Results

### 3.1. LINC01270 Attenuation Upregulates Pro-Inflammatory Cytokines in Lipopolysaccharide (LPS)-Treated THP-1 Cells

Based on previous RNA sequencing conducted in our laboratory examining THP-1 cells stimulated with LPS, several lncRNAs were revealed to be differentially expressed [23]. Out of these, the expression of LINC01270, as reported by The Genotype-Tissue Expression (GTEx) Portal, is highly expressed in the spleen and whole blood (Figure 1A). The functions performed by lncRNAs are often dependent on their cellular localization. Data acquired from the lncBIOATLAS bioinformatics website (http://lncatlas.crg.eu, accessed on 25 March 2022) reveal that the nucleus is apparently its dominant location across various cell lines (Figure 1B), a characteristic seen in most intergenic noncoding RNAs. Similarly, cell fractionation in THP-1 revealed LINC01270 nuclear dominance localization (Appendix A). Accordingly, LINC01270 was chosen as this study’s target. We performed qPCRs on LPS-treated THP-1 cells at different time points to investigate LINC01270 expression and found that LPS upregulates LINC01270 expression 6 and 24 h after stimulation (Figure 1C). The transient decrease observed 8 h post-stimulation suggests that this response follows a biphasic pattern, a characteristic feature of LPS-induced inflammatory activation signals [24,25,26,27].

To explore the role of LINC01270 in monocyte/macrophage-associated inflammatory responses, its expression in THP-1 cells was reduced using a combination of two LINC01270-specific siRNAs (siLINC01270) (Figure 2A). To achieve optimal knockdown efficiency, all experiments were performed 48–72 h post-transfection. Cytokine secretion and expression were examined by ELISA and qPCR in siLINC01270-transfected THP-1 cells treated with LPS. Attenuation of LINC01270 expression upregulated LPS-induced IL-6, IL-8, and MCP-1 mRNA expression (Figure 2B–D) as well as their protein secretion (Figure 2E,F). Although the decrease in LINC01270 levels altered the cytokine secretory profile of THP-1 cells, their phagocytic activity remained unchanged (Appendix A). Collectively, these results indicate that LINC01270 is involved in regulating cytokine expression in THP-1 cells following LPS stimulation.

### 3.2. LINC01270 Knockdown Exacerbates Inflammatory Responses by Upregulating NF-κB

In macrophages, upon LPS binding, Toll-like receptor (TLR)4 triggers a downstream signaling cascade ultimately leading to the onset of inflammatory responses. During these events, the major pro-inflammatory transcription factor NF-κB is activated and induces the transcription of various pro-inflammatory genes [28].

To explore the influence of LINC01270 attenuation on NF-κB activation, alterations in the expression of NF-κB signaling pathway components were ascertained via Western blot. In siLINC01270-transfected THP-1 cells, an upregulation of p65-Ser536 phosphorylation, but not p65-Ser278 phosphorylation, was observed following LPS treatment (Figure 3A,B). NF-κB activation requires phosphorylation and the subsequent degradation of IκB. Compared to the control, siLINC01270 transfection increased IκB phosphorylation. Although siLINC0127 transfection appeared to enhance IκB degradation, this change was not statistically significant. To further clarify the role of LINC01270 in NF-κB regulation, a dual-luciferase reporter assay was performed in HEK293 cells. Transient transfection of siLINC01270, a firefly luciferase construct under a promoter containing NF-κB binding sites, and a CD4–TLR4 expression plasmid exacerbated NF-κB-dependent luciferase activity (Figure 3C,D). CD4–TLR4 constitutively activates the TLR4 signaling pathways [29]. These results substantiate the contention that LINC01270 modulates the LPS-induced inflammatory response in THP-1 cells by regulating NF-κB activation.

### 3.3. LINC01270 Regulates LDOC1 mRNA Stability by Sponging miR-326

lncRNAs exert regulatory functions across diverse cellular processes by sequestering miRNAs, acting as ceRNAs [30]. This miRNA sequestration impedes the interaction between miRNA and target mRNAs, ultimately attenuating their regulatory impact on gene expression.

In pursuit of elucidating the mechanism by which LINC01270 regulates NF-κB-mediated inflammation in LPS-treated THP-1 cells, we sought to determine the miRNA-mRNA pairs possibly affecting NF-κB activation. Leucine zipper downregulated in cancer 1 (LDOC1) was of particular interest since LDOC1 has been shown to impose an inhibitory effect on NF-κB activity [20,21,22]. Using the LncRNASNP database, miR-326 was predicted to interact with LDOC1.

Further, the miR-326–mRNA interactome profile, based on two different miRNA–mRNA interaction databases, Targetscan and miRDB, nominated LDOC1 mRNA as one of the most probable targets of miR-326 (Figure 4A,B). In addition, previous research has already established the liaison between miR-326 and LINC01270 [12,13].

LPS treatment upregulated LINC01270 expression in THP-1 cells (Figure 1C). Given the role of lncRNAs in sequestring miRNAs, the expression profile of miR-326 was also examined under the same conditions. As expected, LPS treatment downregulated miR-326 levels in THP-1 cells (Figure 4C). Additionally, LDOC1 expression was reduced in siLINC01270-transfected THP-1 cells following LPS treatment (Figure 4D). Consistently, LINC0120 knockdown led to an increase in miR-326 levels (Appendix A), and a decoy RNA containing the miR-326 binding site on LINC01270 effectively lowered miR-326 levels, further supporting the LINC01270-miR-326 interaction. Furthermore, treatment with miR-326 inhibitor, a chemically modified antisense oligonucleotide, effectively elevated both LDOC1 mRNA and LINC01270 RNA levels (Figure 4E,F).

To further demonstrate the role of miR-326 in mediating LINC01270’s effect on LPS-induced inflammatory responses, miR-326 inhibitor was co-transfected with siLINC01270. The miR-326 inhibitor counteracted the decrease in LDOC1 mRNA and protein levels due to siLINC01270 in THP-1 cells (Figure 4G–I). Moreover, LDOC1 3′UTR luciferase reporter results confirm the binding between miR-326 and LDOC1 mRNA (Appendix A). Additionally, the NF-κB-mediated enhancement of luciferase reporter expression in siLINC01270-transfected HEK293 cells was efficiently reversed by miR-326 inhibitor (Figure 4J). In THP-1 cells, suppression of miR-326 also blocked siLINC0127-mediated upregulation of p65-Ser536 and IκB phosphorylation levels (Figure 4K,L) and IL-6, IL-8, and MCP-1 mRNA expression (Figure 4M–O). These findings underscore miR-326’s importance as a pivotal component of the regulatory network associated with LINC01270 during the inflammatory response elicited by LPS and implicate the targeting of LDOC1 as its primary mechanism.

### 3.4. A Synthetic RNA Fragment Mimicking the miR-326 Binding Site of LINC01270 Reverses the Effect of siLINC01270 on LPS-Induced Inflammation in THP-1 Cells

Decoy RNAs, also known as competitive inhibitors or competitive decoys, are synthetic RNA molecules designed to competitively bind specific RNA-interacting molecules, such as miRNAs or lncRNAs, to modulate gene expression or cellular processes [31,32].

To further validate the interrelationships among LINC01270, miR-326, and LDOC1 mRNA, a decoy RNA representing the putative binding site of miR-326 was synthesized (Figure 4A, Table 3). Transfection of THP-1 with the decoy RNA alone upregulated both LINC01270 RNA and LDOC1 mRNA transcript levels (Figure 5A,B). Decoy fragment transfection also effectively counteracted the decrease in LDOC1 mRNA levels seen in siLINC01270-transfected THP-1 cells (Figure 5C). Similar to the miR-326 inhibitor, the decoy RNA neutralized the siLINC01270-mediated enhancement of NF-κB-luciferase activity in HEK293 cells (Figure 5D) and the phosphorylation of p65-Ser536 and IκB (Figure 5E,F) and pro-inflammatory cytokine expression in THP-1 cells (Figure 5G–I). These results confirm the implication of the LINC01270–miR-326–LDOC1 axis in the regulation of the NF-κB-mediated inflammatory response to LPS in THP-1 cells.

### 3.5. LINC01270 Overexpression Suppresses NF-κB Activation in HEK293 Cells

Demonstrating the consequences of LINC01270 attenuation during the onset of the inflammatory response prompted us to study the effects of LINC01270 overexpression. To express LINC01270 ectopically, we used the CRISPR-activation method. HEK293 cells were transiently transfected with a plasmid expressing dCas9 fused with VP64 transcriptional activator together with the tracrRNA–crRNA complex (gRNA) targeting the LINC01270 transcription starting site. LINC01270 expression increased moderately after transfection of the gRNA (Figure 6A). The overexpression of LINC01270 in HEK293 cells downregulated miR-326 expression, probably due to an increased sponging effect of LINC01270 (Figure 6B) and, consequently, upregulated LDOC1 mRNA levels (Figure 6C). This increase in LDOC1 mRNA presumably suppresses TLR4-mediated NF-κB activation, which was confirmed by dual luciferase assay (Figure 6D). Similarly, the abundance of LINC01270, via its effects on miR-326/LDOC1 interactions, downregulated TNF-α-induced NF-κB activation by downregulating the phosphorylation of both P65-Ser536 and IκB (Figure 6E,F).

## 4. Discussion

The advancement of high-throughput sequencing technologies has facilitated the identification of numerous lncRNAs. Nonetheless, elucidating the functional roles of the majority of these non-coding RNAs in cellular processes remains a challenge. The growing interest in lncRNA functions has mainly focused on their involvement in various aspects of cancer progression, including their potential utility as prognostic or diagnostic biomarkers [33]. However, despite the pivotal role of inflammation and immune cell responses in the initiation and advancement of cancer, the amount of research on the involvement of lncRNAs in the regulation of these processes remains relatively low. In fact, the inappropriate regulation of the inflammatory response has detrimental effects. For instance, chronic inflammation can promote carcinogenesis and metastasis [34], Inflammation can also affect the host’s immune response to tumors; thus, it can be targeted in cancer immunotherapy to enhance the effectiveness of chemotherapy [35]. Immune cells also exert significant influence over the fate of malignant cells by secreting a diverse array of cytokines, which can either promote cancer cell survival and proliferation or trigger their apoptosis [36]. Therefore, exploring the function of lncRNAs in inflammation regulation might be the key to a better understanding of cancer progression as well as other inflammatory diseases such as sepsis and rheumatoid arthritis.

In this study, we sought to shed light on the potential impact of LINC01270 on the inflammatory response and demonstrate, for the first time, the involvement of LINC01270 in regulating the pro-inflammatory response in the monocytic leukemia cell line THP-1 stimulated with LPS. The upregulation of LINC01270 expression by LPS incited us to speculate that LINC01270 might impose a negative modulatory influence. Our findings demonstrate that LINC01270 attenuates NF-κB activation by regulating the expression of LDOC1 through the sequestration of miR-326 (Figure 7). Previous reports on miR-326’s effects on TET2 revealed its pro-inflammatory nature [18]. However, an anti-inflammatory role of miR-326 via the destabilization of TLR4 mRNA has also been reported [19]. The disparity in miR-326’s function in modulating NF-κB activity can be attributed to the aptitude of miRNAs to target multiple genes [37,38]. Additionally, an miRNA might have different targets in different cells depending on their abundance, leading to distinct functional outcomes [39,40]. Moreover, different cellular environments, for example, the presence of various regulatory molecules and signaling pathways depending on the cell type, produce different miRNA outcomes. Additionally, miRNAs are integral components of a vast gene regulatory network. While miRNAs regulate mRNA stability and translation, they are themselves regulated by various cellular components, including transcription factors, DNA methylation, histone modifications, RNA-binding proteins, and lncRNAs [41,42,43,44].

We observed that, in LPS-induced THP-1 cells, the increase in LINC01270 is associated with a decrease in miR-326 levels, suggesting that miR-326 is sequestered by LINC01270. The attenuation of LINC01270 expression thus frees miR-326, allowing its interaction with the putative binding site located in LDOC1 mRNA’s 3′UTR region to mitigate its downstream effects. Interestingly, the mechanism by which LDOC1 regulates NF-κB is not yet well established. The few studies on this matter show disparities in LDOC1’s mode of action. For instance, Zhao et al. [22] illustrated the ability of LDOC1 to downregulate P65 nuclear content and inhibit the degradation of IκB, while, on the other hand, a study carried out by Thoompumkal et al. [45] highlighted LDOC1’s ability to interact with and destabilize guanine nucleotide binding protein-like 3-like (GNL3L). In cancer conditions, LDOC1 expression is downregulated and GNL3L ultimately enhances NF-κB activity by regulating phosphorylation levels and stability of the NF-κB p65 subunit and IκB [45]. The activation of NF-κB is a tightly regulated process, primarily controlled by the IκBα kinase (IKK)-mediated phosphorylation of IκB. Normally, IκB sequesters NF-κB, preventing its entry into the nucleus. The phosphorylation of IκB promotes the downstream phosphorylation of the p65 subunit at different residues, enabling the nuclear translocation of NF-κB, which then mediates the transcription of inflammatory genes [46]. Our results align closely with these findings.

In spite of the consistency between our findings and previous conceptions of LDOC1’s function in the inflammatory response, the outcome of our investigation cannot affirm decisively whether LDOC1 modulates the phosphorylation of p65 and IκB via its interaction with GNL3L or other regulatory proteins. Although LDOC1 is predominantly localized in the nucleus [47], our results and previous findings suggest its action in the cytosol. In addition, shifts in the subcellular localization of LDOC1 have been observed in PTC cells [48].

Furthermore, the presence of a leucine zipper motif and SH3 domain within its structure grants LDOC1 the ability to bind DNA and interact with various proteins, respectively, and suggests LDOC1’s possible involvement in a variety of processes. Other than NF-κB, LDOC1 is involved in regulating facets of inflammation through the regulation of different transcription factors and pathways. LDOC1 interacts with Ligand-of-Numb protein X1 (LNX1), an E3 ligase, to induce Janus kinase 2 (JAK2) ubiquitination and degradation, which negatively regulates STAT3 activation and IL-6 secretion in lung cancer cells [49]. In oral squamous cell carcinoma, LDOC1 inhibited microbe-induced IL-1β production by regulating the phosphoinositide 3-kinases (PI3K)/protein kinase B (Akt)/phospho-glycogen synthase kinase 3 beta (pGSK-3β) signaling pathway [50]. The multifaceted impact of LDOC1, along with the diverse interactions of LINC01270 and miR-326 with other effectors, prompts us to further explore the role of LINC01270 in modulating inflammation through alternative pathways, as well as its potential effects on other cellular processes.

While our findings focus on the functional interactions within the LINC01270/miR-326/LDOC1 axis, we acknowledge the possibility that LINC01270 may also regulate the transcription of LDOC1 or miR-326. Although our experimental evidence suggests otherwise, we cannot conclusively exclude this potential mechanism. Future studies employing more targeted approaches, such as chromatin immunoprecipitation (ChIP) assays or promoter activity analyses, are warranted to investigate whether LINC01270 directly influences the transcriptional regulation of these genes. This exploration would further refine our understanding of the regulatory dynamics within this axis.

## 5. Conclusions

The current study investigated the role of the lncRNA LINC01270 in modulating the inflammatory response. LPS treatment upregulated LINC01270 in the human monocytic leukemia cell line THP-1, and its suppression via siRNA enhanced NF-κB activity and the production of pro-inflammatory cytokines IL-6, IL-8, and MCP-1. The knockdown of LINC01270 also led to the downregulation of LDOC1, a known suppressor of NF-κB. Through bioinformatics and experimental validation, miR-326 was identified as a mediator in this regulatory pathway. Suppression of LINC01270 frees miR-326 to bind to LDOC1 mRNA, reducing its stability and leading to enhanced NF-κB activation and inflammation. Additionally, synthetic RNA agents that disrupt the interaction between LINC01270, miR-326, and LDOC1 mitigated the inflammatory response. This study confirms that LINC01270 functions as a ceRNA, sponging miR-326 to maintain LDOC1 levels and consequently suppress NF-κB activation. Overexpression of LINC01270 resulted in reduced NF-κB activity and inflammatory cytokine production, further supporting its role as a negative regulator of inflammation. Thus, this study provides insight into LINC01270’s role in modulating inflammatory responses to LPS stimulation in THP-1 cells via the miR-326–LDOC1 axis, which negatively regulates NF-κB. Further, these results can be applied to the development of therapeutic methods for the various diseases in which NF-κB-mediated inflammatory activation plays a major role in the pathogenesis.

## Figures and Tables

**Figure 1 cells-13-02027-f001:**
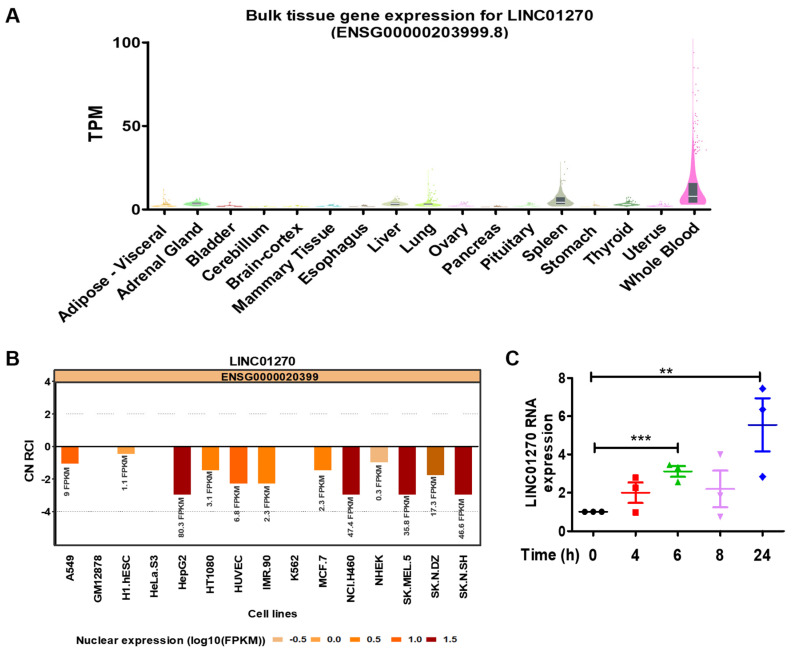
LPS treatment upregulates LINC01270 expression in THP-1 cells. (**A**) Bulk tissue gene expression for LINC01270 (GTEx portal). (**B**) Subcellular localization of LINC01270 analyzed by the bioinformatics website (http://lncatlas.crg.eu, accessed on 25 March 2022). (**C**) THP-1 cells were treated with 1 µg/mL LPS for the indicated time points, and LINC01270 RNA levels were quantified using qRT-PCR. Relative RNA levels are presented as fold-change in comparison to 0 h. Data are presented as mean ± SEM (n = 3). ** *p* < 0.01, *** *p* < 0.001.

**Figure 2 cells-13-02027-f002:**
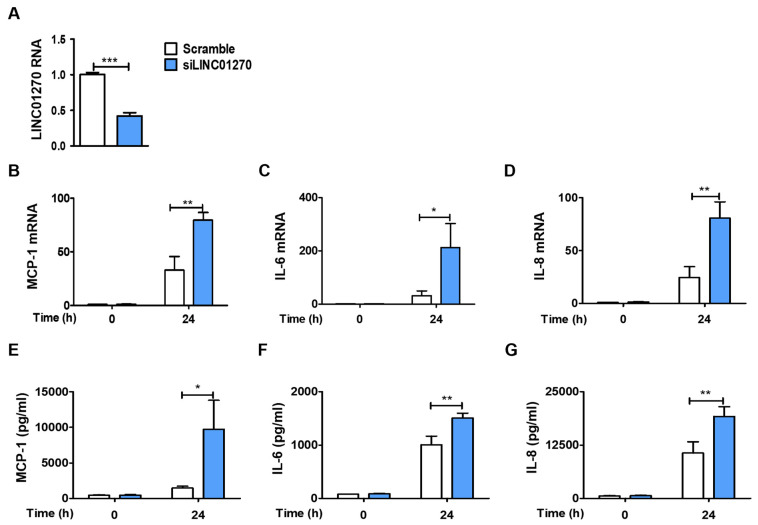
LINC01270 attenuation upregulates pro-inflammatory cytokines in THP-1 cells. (**A**) Cells were transfected with either scramble or siLINC01270 for 48 h, and knockdown efficiency was measured via qRT-PCR. (**B**–**D**) Transfected THP-1 cells were stimulated with 1 µg/mL LPS for 0 and 24 h, and mRNA levels of MPC-1, IL-6 and IL-8 were assessed via qRT-PCR. (**E**–**G**) Secretion levels of MPC-1, IL-6 and IL-8 were quantified using ELISA after treating transfected THP-1 cells with 1 µg/mL LPS. Data are presented as mean ± SEM (n = 3). * *p* < 0.05, ** *p* < 0.01, *** *p* < 0.001.

**Figure 3 cells-13-02027-f003:**
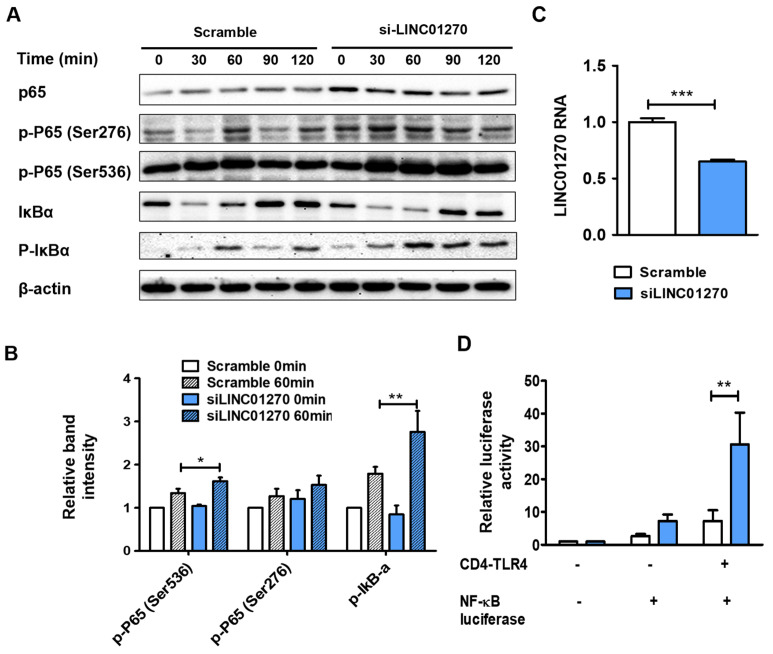
LINC01270 knockdown mediates inflammatory response exacerbation by upregulating NF-κB. (**A**,**B**) THP-1 cells transfected with siLINC01270 were stimulated with 1 µg/mL of LPS for the indicated time periods. The cell lysates were subjected to Western blot. Specific antibodies against p65, phospho-p65 (Ser276 and Ser536), IκBα, phospho–IκBα, and β-actin were used to detect the respective proteins. The relative intensities of the bands were quantified through densitometry, with β-actin bands serving as the internal control. (**C**) HEK293 cells were transfected with siLINC01270 for 48 h, and knockdown efficiency was measured via qRT-PCR. (**D**) HEK293 cells transfected with either scramble or siLINC01270 were co-transfected with an NF-κB luciferase construct and a CD4-TLR4 construct. A Renilla luciferase construct was used as an internal control to which the expression of the firefly luciferase reporter gene was normalized. Data are presented as mean ± SEM (n = 3). * *p* < 0.05, ** *p* < 0.01, *** *p* < 0.001.

**Figure 4 cells-13-02027-f004:**
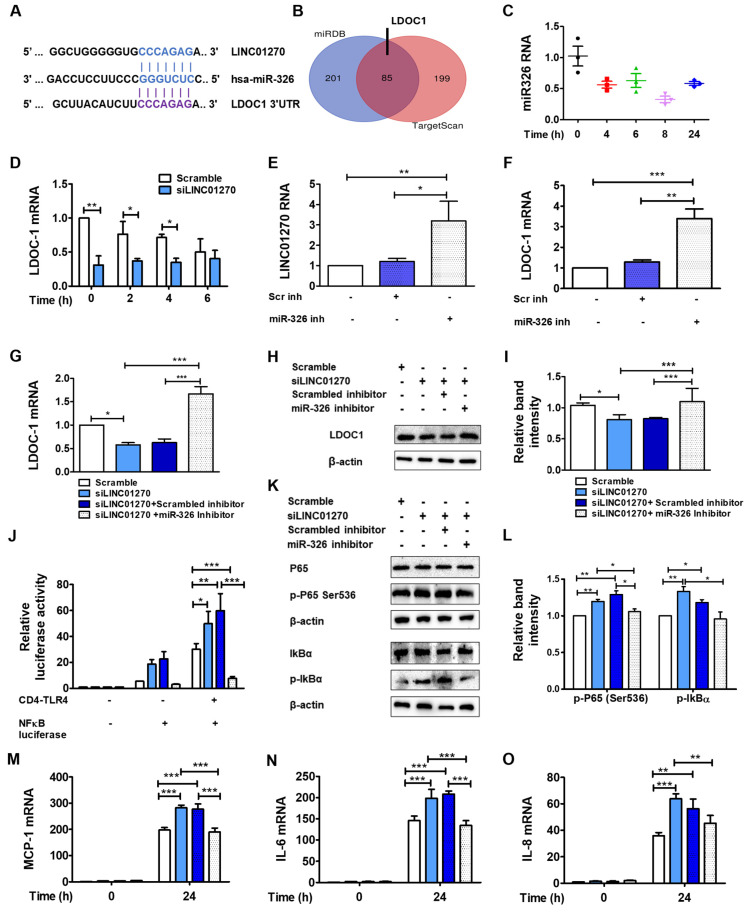
LINC01270 sponges miR-326 targeting LDOC1 to modulate LPS-induced inflammatory response in THP-1 cells. (**A**) Predicted interactions between LINC01270 and miR-326 and between miR-326 and the 3′UTR region of LDOC1. (**B**) Prediction of miR-326 downstream protein targets shown in a Venn diagram. (**C**) miR-326 expression levels measured by qRT-PCR in THP-1 cells treated with 1 µg/mL LPS. (**D**) LDOC1 mRNA expression levels measured by qRT-PCR in siLINC01270-transfected THP-1 after stimulation with 1 µg/mL LPS for the indicated timepoints. (**E**,**F**) THP-1 cells were transfected with either scrambled or miR-326 inhibitor for 48 h; LINC01270 and LDOC1 expression levels were determined by qRT-PCR. (**G**) LDOC1 expression measured by qRT-PCR in THP-1 cells co-transfected with siLINC01270 and miR-326 inhibitor and treated with 1 µg/mL LPS for 1 h. (**H**,**I**) LDOC1 expression measured by Western blot after LPS treatment for 1 h at 1 µg/mL. Relative band intensity was quantified by densitometry, with β-actin bands serving as the internal control. (**J**) HEK293 cells were co-transfected with siLINC01270 and miR-326 inhibitor, and NF-κB activation was measured by a dual luciferase assay. (**K**,**L**) Western blot analysis of NF-κB cascade proteins in co-transfected THP-1, and band intensities were quantified through densitometry. (**M**–**O**) Cytokine expression measured by qRT-PCR in THP-1 cells co-transfected with siLINC01270 and miR-326 inhibitor treated with 1 µg/mL of LPS for 24 h. Data are presented as mean ± SEM (n = 3). * *p* < 0.05, ** *p* < 0.01, *** *p* < 0.001.

**Figure 5 cells-13-02027-f005:**
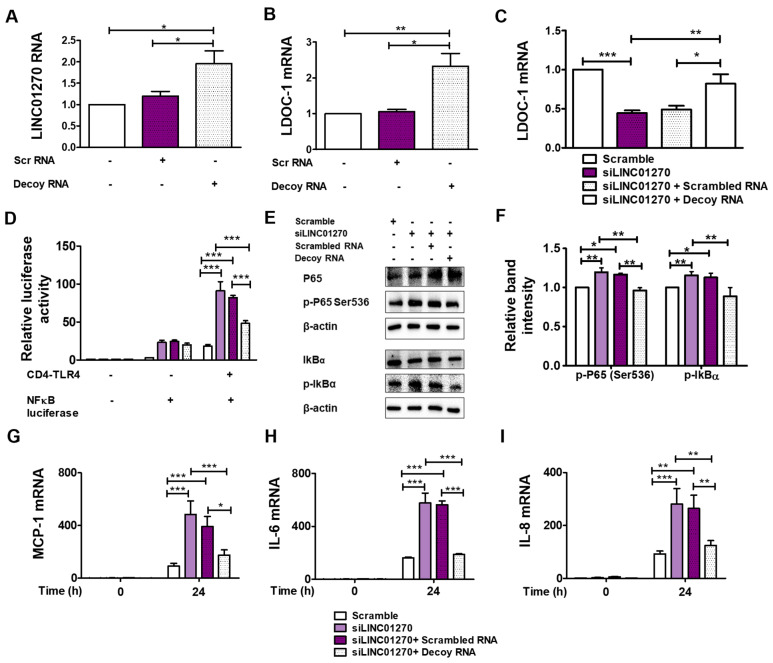
LINC01270 decoy fragment mimicking the miR-326 binding site reverses the effect of LINC01270 downregulation on LPS-induced inflammation. (**A**,**B**) THP-1 cells were transfected with either scrambled control or decoy RNA for 48 h, and LINC01270 and LDOC1 expression levels were determined by qRT-PCR. (**C**) LDOC1 expression measured by qRT-PCR in THP-1 cells co-transfected with siLINC01270 and decoy RNA and stimulated with LPS (1 µg/mL, 1 h). (**D**) HEK293 cells were co-transfected with siLINC01270 and decoy RNA, and NF-κB activation was measured by a dual luciferase assay. (**E**,**F**) NF-κB cascade protein expression levels after LPS treatment (1 µg/mL, 1 h) were measured by Western blot. Relative band intensity was quantified by densitometry. (**G**–**I**) Cytokine expression levels were measured by qRT-PCR in THP-1 cells co-transfected with siLINC01270 and decoy RNA and treated with 1 µg/mL LPS for 24 h. Data are presented as mean ± SEM (n = 3). * *p* < 0.05, ** *p* < 0.01, *** *p* < 0.001.

**Figure 6 cells-13-02027-f006:**
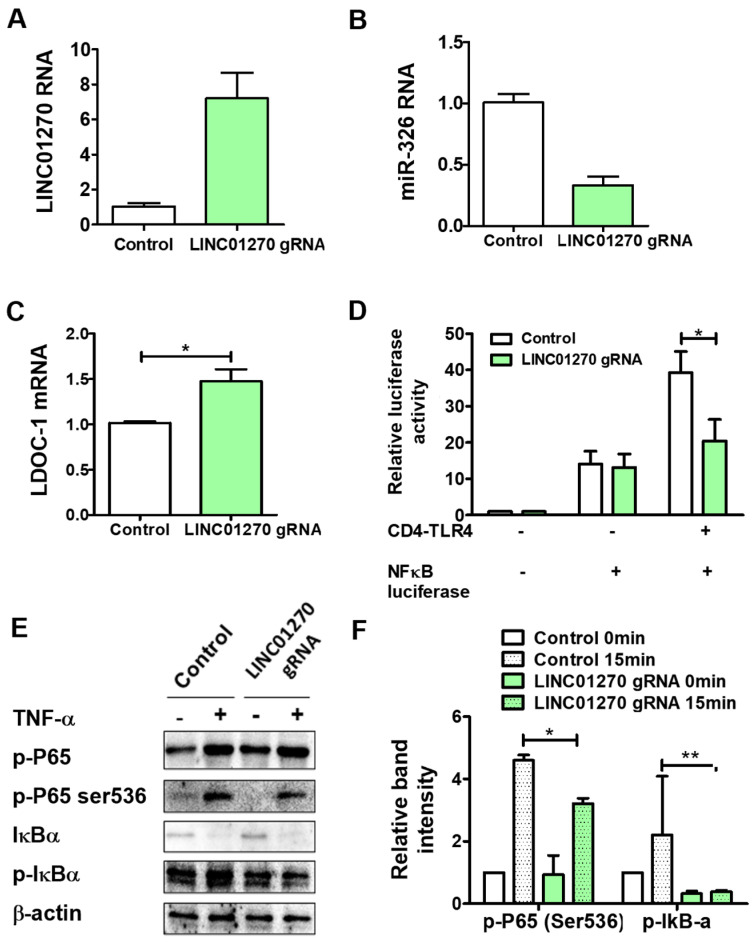
LINC01270 overexpression attenuates NF-κB activation in HEK293 cells. (**A**) Cells were transfected with dCas9-VP64 vector alone (control) or with LINC01270 crRNA–TracrRNA (gRNA) for 48 h; LINC01270 expression levels were measured by qRT-PCR. (**B**,**C**) miR-326 and LDOC1 expression levels were measured using qRT-PCR in HEK293 cells transfected as in (**A**). (**D**) NF-κB luciferase activity was measured in HEK293 cells overexpressing LINC01270 via a dual luciferase assay. (**E**,**F**) Western blot analysis of NF-κB cascade proteins in HEK293 cells treated with 100 ng/mL TNF-α for 1 h. Relative band intensities were quantified through densitometry. Data are presented as mean ± SEM (n = 3). * *p* < 0.05, ** *p* < 0.01.

**Figure 7 cells-13-02027-f007:**
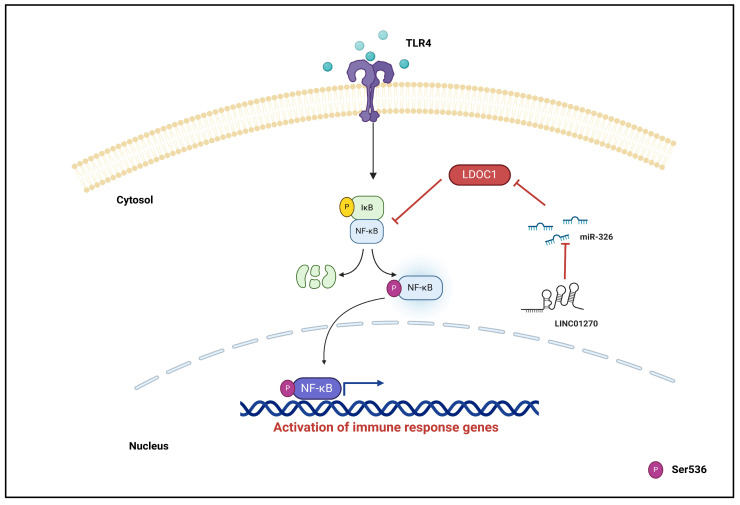
Mechanism diagram of LINC01270 negatively regulating NF-κB signaling through the miR-326-LDOC1-axis. This diagram illustrates how LINC01270 acts as a sponge for miR-326, preventing miR-326 from binding to the 3′UTR of LDOC1 mRNA. By sequestering miR-326, LINC01270 upregulates LDOC1 expression. Increased LDOC1 levels inhibit NF-κB signaling, leading to reduced expression of NF-κB target genes, which are involved in the inflammatory response. Consequently, the overall effect of LINC01270 is the attenuation of NF-κB-mediated inflammation.

**Table 1 cells-13-02027-t001:** qPCR primers used in this study.

Name	Sequence (5′ → 3′)	
β-ACTIN	Forward	TGA GAT GCG TTG TTA CAG GAA GTC
	Reverse	GAC TGG GCC ATT CTC CTT AGA GA
IL-6	Forward	GGT ACA TCC TCG ACG GCA TCT
	Reverse	TTG ATT GCA TCT GGC TGA GC
IL-8	Forward	ATA AAG ACA TAC TCC AAA CCT TTC CAC
	Reverse	AAG CTT TAC AAT AAT TTC TGT GTT GGC
MCP-1	Forward	ACT CTC GCC TCC AGC ATG AA
	Reverse	TTG ATT GCA TCT GGC TGA GC
LINC01270	Forward	CGACGCTGTCTCAGACTCTC
	Reverse	GTGCTGCAGCTCTATAGGACA
LDOC1	Forward	GAA CCG ATT CTG CAA CGA CG
	Reverse	ACT GTT TCA TCT CAT CGA GGA

**Table 2 cells-13-02027-t002:** Primary antibodies used for the Western blots in this study.

Antibody Name (Catalog Number)	Company	Dilution
LDOC1 (sc-81103)	Santa Cruz Biotechnology	1:200
NF-κB p65 (F-6) (sc-8008)	Santa Cruz Biotechnology	1:1000
β-Actin (sc-47778)	Santa Cruz Biotechnology	1:1000
NF-κB p65 (Ser276) (sc-101749)	Santa Cruz Biotechnology	1:1000
NF-κB p65 (Ser536) (93H1) (3033S)	Cell Signaling	1:1000
IκBα (9242S)	Cell Signaling	1:1000
phosphorylated IκBα (S32/36) (9246S)	Cell Signaling	1:500

**Table 3 cells-13-02027-t003:** Sequences of the siRNAs, decoy RNA, and gRNA used in this study.

Name	Sequence (5′ → 3′)	
LINC01270 siRNA 1	Sense	GGGUCACAUAGCGAGAGUUAU
	Antisense	AUAACUCUCGCUAUGUGACCC
LINC01270 siRNA 2	Sense	GGUCACAUAGCGAGAGUUAUU
	Antisense	AAUAACUCUCGCUAUGUGACC
Decoy fragment		UUCCCAGAGAA
LINC01270 crRNA		CUGAACUGGCUGACCCCUUC

## Data Availability

The original contributions presented in this study are included in the article/Appendix A. Further inquiries can be directed to the corresponding author.

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
