# Peer review of "LINC01270 Regulates the NF-κB-Mediated Pro-Inflammatory Response via the miR-326/LDOC1 Axis in THP-1 Cells"

_cells, 2024, doi:10.3390/cells13232027_

Round 1
Reviewer 1 Report (Previous Reviewer 1)
Comments and Suggestions for Authors
I commend the authors for considering my suggestions and comments and responding appropriately. The revised manuscript is significantly improved compared to the first version and is now acceptable for publication. However, the authors have not fully addressed my concern regarding whether LINC01270 regulates the transcription of LDOC1 and miR-326.
Minor concerns:
- There are two Figure S2 labels. The second one should be corrected to Figure S3.
- Total p65 and IκBα should be included in Figure 5E for completeness.
Author Response
We sincerely appreciate the reviewer’s invaluable comments, which have significantly enhanced the clarity and focus of our manuscript. Below, we provide detailed responses to each comment and outline the corresponding revisions.
The reviewer has noted that the question regarding LINC01270's regulation of LDOC1 and miR-326 transcription remains unresolved. While we acknowledge the potential for LINC01270 to regulate the transcription of LDOC1 or miR-326, our manuscript primarily focuses on elucidating the functionality of the LINC01270/miR-326/LDOC1 axis. Thus, while we do not exclude the possibility of transcriptional regulation, it is not the central scope of this study.
Regarding this issue, we would like to clarify that loss- or gain-of-function experiments followed by qPCR would not conclusively determine whether the observed changes in LDOC1 and miR-326 expression levels are due to transcriptional repression or the direct binding interactions among LINC01270, miR-326, and LDOC1. Instead, our focus is on the LINC01270/miR-326/LDOC1 axis, supported by the following evidence:
- Functional Evidence Using Luciferase Assay: We utilized a luciferase vector containing the 3' untranslated region (3'UTR) of LDOC1 to demonstrate that miR-326 inhibition increases the relative luciferase activity of LDOC1 3'UTR (Figure S4A-B).
- Binding Evidence: Prior studies have demonstrated the interaction between LINC01270 and miR-326 through dual-luciferase assays (PMID: 36694453, PMID: 35345980).
- Experimental Observations: If LINC01270 directly regulated LDOC1 transcription, it is unlikely that treatments with a miR-326 inhibitor or decoy RNA would restore LDOC1 mRNA levels following LINC01270 knockdown.
To address the reviewer’s concerns more comprehensively, we have added a statement to the Discussion section (lines 458-465) acknowledging the possibility of LINC01270-mediated transcriptional regulation of LDOC1 or miR-326 and highlighting the need for future studies to explore this aspect in more detail.
Minor Revisions:
- We have corrected the labeling of Figure S2 to Figure S3 in the supplementary data file.
- Western blot images for p65 and IκBα have been included in Figures 4K, 5E, and 6E as requested.
We hope these clarifications and revisions address the reviewer’s concerns effectively. Thank you for your thoughtful feedback and for providing us with the opportunity to further improve our manuscript.
Reviewer 2 Report (Previous Reviewer 3)
Comments and Suggestions for Authors
The authors have successfully addressed the raised issues and increased the quality of their manuscript.
Author Response
Thank you for your review and suggestions.
This manuscript is a resubmission of an earlier submission. The following is a list of the peer review reports and author responses from that submission.
Round 1
Reviewer 1 Report
Comments and Suggestions for Authors
In this manuscript, Arab et al. explored the function of LINC01270 in the LPS-triggered immune response in THP-1 cells. They provided evidence and concluded that LINC01270 likely regulates the NF-kB mediated pro-inflammatory response via the miR-326/LDOC1 axis. However, based on the existing evidence in the manuscript, I believe the conclusion is not strongly supported and does not present a significant conceptual advance. Therefore, I do not consider it a strong candidate for publication in Cells as it stands.
Some specific concerns:
- Figure 1A: LINC01270 is highly expressed in the spleen and whole blood but is not upregulated in these tissues.
- Figure 1: The authors should provide experimental validation of the localization of LINC01270 in THP-1 cells, which is the cell line used in this project.
- Figure 1C: The expression level of LINC01270 decreases at 8h after treatment, compared to 6h. Is there any explanation for this?
- For nuclear lncRNA, siRNA often shows lower knockdown efficiency. The authors should also perform ASO treatment, which is widely used for nuclear lncRNA knockdown.
- What is the absolute abundance of LINC01270 and miR-326 in THP-1 cells? It requires a very high level of expression for lncRNA to function as a miRNA sponge. I am questioning whether the expression of LINC01270 is sufficiently high to act as a miRNA sponge.
- The authors should provide evidence showing the transcription level of miR-326 in Figure 4 to exclude the possibility that the downregulation of miR-326 after treatment is not a result of reduced transcription.
- What is the internal control in Figure 4C?
- The authors provide evidence that miR-326 could regulate the expression of LINC01270 and LDOC1, but this is not enough to support the conclusion. For instance, there is no evidence to exclude the possibility of a direct regulatory relationship between these two genes. Since LINC01270 is a nuclear RNA (as claimed in the paper), does LINC01270 regulate the transcription of LDOC1 and miR-326? Does decoy RNA change the level of miR-326?
- I cannot understand Figures 5A and 5B. Transfection of decoy RNA alone did not change the expression of LINC01270 and LDOC1 (the second bar), and co-transfection of decoy RNA and scramble RNA enhanced the expression of both (the third bar), which is inconsistent with the context. How can this be explained?
- To be honest, I cannot discern the differences between different samples in Figures 5E and 5F. Additionally, there is no loading control in Figure 6E.
Reviewer 2 Report
Comments and Suggestions for Authors
The manuscript describes for the first time the regulation of NF-κB by the ceRNA around LINC01270 which includes miR-326 and LDOC1.
General Comments.
The authors smartly explore the inflammatory response by focusing on a hot topic such as the complex role of ncRNAs. This study sheds light on the role of LINC01270 as regulator of NF-κB activation and its effects mediated by miR-326 and LDOC1 mRNA by in vitro models in order to confirm the potential interactions among them.
The manuscript is well redacted; experiments are well designed, fully explained and contextualized. However, and due to the complexity of the mechanisms of action of ncRNAs within a ceRNA, a brief description about their role and the main actions of each type of ncRNA would make the results easier to understand.
Minor points:
- - The authors have used THP-1 and HEK293 cell lines in order to perform the in vitro experiments. Why the transfection of siLINC01270, miR-326 and their decoy RNAs was done in THP-1 cells and the luciferase assays were ran in HEK293? Please, provide the information about the criteria followed in order to determine the appropriated cell line in each kind of experiments.
- - First time LDOC1 is mentioned in the text is in line 261, please include the fully name and a sentence aiming to contextualize it a little bit more due to its relevance in this study.
- -Line 273-274: “Moreover, the usage of miR-326 inhibitor effectively increased LDOC1 mRNA and LINC01270 RNA levels”, what is “miR-326 inhibitor” meaning? Are the authors using miR-326 mimics in other to confirm if the experiments have been performed by using mimics or inhibitors? Please, clarify it all around the text.
Reviewer 3 Report
Comments and Suggestions for Authors
Arab and colleagues present a paper aimed at describing the LINC01270/miR-326/LDOC1 axis and its effect on NF-κB pro-inflammatory regulation. The experimental strategy is clearly laid out and the presented results are sound. However, there are a few aspects that the authors must improve before publication.
Major points
The authors must provide experimental evidence of the miR-326/LDOC1 interaction to complement that of the LINC01270/miR-326 integration presented in Reference 11 and thus ascertain the LINC01270/miR-326/LDOC1 regulation axis. Consider using a luciferase reporter assay with the LDOC1 3’-UTR region.
Figure 2A shows the decrease in LINC01270 expression after 48h, yet expression and secretion of cytokines were assayed at 24 h. Please clarify.
Figure 4D shows a decrease over time in LDOC1 expression when cells are transfected with a scrambled siRNA. Please explain
The authors state that ‘the usage of miR-326 inhibitor effectively increased LDOC1 mRNA and LINC01270 RNA levels’ (Lines 273-274), but Fig 4D and 4E show increased LDOC1 and LINC01270 only when transfected with both the specific and the scrambled inhibitors
It is hard to agree with the authors on their interpretation of the band intensities in some Western blot assays based on the provided figures: In Figure 3A, P-IκBα detection at 0 min is barely visible in the scrambled, 0-min column and is slightly more evident in the si-LINC01270, 0-min column; so the increase at 60 min seems greater in the scrambled group. Yet, the graph in Figure 3B reflects the opposite. In the same figure, the variation in IκBα detection is not discussed. Similarly, the four P-IκBα bands in Figures 4K and 5E seem very similar, while the authors claim that they are significantly different. The IκBα bands in each of these figures seem uniform, but Figure 5E shows an overall increase that is not discussed either. Perhaps there are images that better represent the results among the performed experimental repeats.
Overall, the results show that LINC01270 attenuation upregulates pro-inflammatory cytokines and upregulates NF-κB. Most likely, this regulation happens through the LINC01270/miR-326/LDOC1 regulation axis (see comment above). Given that the mechanism through which LDOC1 phosphorylates IκBα is not established yet –as the authors acknowledge in the Discussion section–, and that the evidence supporting this particular phosphorylation is not conclusive (see comments above), why place so much emphasis on it?
Minor points
Please provide catalog numbers for the antibodies used
Consider using the adjective form ‘scrambled’ throughout the manuscript, especially when combining it with nouns such as scrambled inhibitor.
Line-71 The term sponge does not usually apply to miRNAs themselves, but to ssRNAs that ‘capture’ them (See DOI: 10.1261/rna.2414110)